# Body Weight Effects on Extra-Osseous Subtalar Arthroereisis

**DOI:** 10.3390/jcm8091273

**Published:** 2019-08-22

**Authors:** Chiun-Hua Hsieh, Chia-Che Lee, Tzu-Hao Tseng, Kuan-Wen Wu, Jia-Feng Chang, Ting-Ming Wang

**Affiliations:** 1Department of Orthopedic Surgery, National Taiwan University Hospital, Taipei City 100, Taiwan; 2Department of Internal Medicine, Shuang Ho Hospital, Taipei Medical University, New Taipei City 235, Taiwan

**Keywords:** body mass index, extra-osseous subtalar arthroereisis, implant extrusion, pediatric flexible flatfoot

## Abstract

Implant extrusion in subtalar arthroereisis is a common complication for pediatric flexible flatfoot. However, there were a limited number of articles addressing the body weight effects on implant extrusion after the procedure. We conducted a 24-month follow-up assessment after subtalar arthroereisis. Surgical patients who underwent the Vulpius procedure were retrospectively collected from May 2010 to January 2017, including 59 cases of both feet having implants in situ and 43 cases of both feet having implant extrusion. The average age of 102 patients was 9 years old. The mean body mass index (BMI) of the implant in situ group was 19.5, whilst the extrusion group was 21.2 (*p* = 0.035). The inter-observer correlation was excellent. There were 11 cases (39.3%) of bilateral extrusion in the overweight group (BMI ≥ 24) and 13 cases (23.2%) in the low body weight group (BMI ≤ 18.5) (*p* < 0.0004). Postoperative radiographic angles were corrected in both the implant in situ group and the extrusion group. Nonetheless, the implant in situ group revealed better postoperative outcomes of Meary’s angle and the talonavicular angle from an anterior-posterior view, and the talar inclination angle from a lateral view. We conclude that a higher BMI is related to implant extrusion and worse results after subtalar arthroereisis. Further prospective study to investigate whether preoperative weight loss results in improved surgical outcomes is warranted in the future.

## 1. Introduction

Subtalar arthroereisis is a prime procedure for pediatric flexible flatfoot, and implant extrusion is a common complication during the postoperative follow-up period. As described by Bernasconi et al. [1], sinus tarsi pain is the most frequent complaint. However, a limited number of articles addressing implant extrusion have been published. We hypothesized that implant extrusion is associated with a higher body mass index (BMI), diminishing the effect of flatfoot correction. Thus, a retrospective study was conducted to identify the correlation between BMI and implant extrusion.

The aims of our study were to investigate (1) the implant extrusion rate after extra-osseous subtalar arthroereisis in pediatric flexible flatfoot after a 24-month follow-up; (2) whether a higher BMI is relevant to implant extrusion and worse surgical outcomes.

## 2. Experimental Section

### 2.1. Patient Criteria

We retrospectively reviewed children with bilateral flexible flatfoot who underwent bilateral extra articular subtalar arthroereisis (Bioarch^®^, Wright) with a bilateral Vulpius procedure from May 2010 to January 2017. The indication of surgery is symptomatic flexible flatfoot with heel cord tightness in children at the age of 7 to 11 years old. The symptoms included plantar pain during activity, heel pain during or after sports, or frequent muscle fatigue after long standing. We excluded the feet with accessory navicular bones. All study patients received conservative treatment including calf stretching exercises, insoles, and non-steroid anti-inflammatory medication for at least 6 months before operation.

### 2.2. Preoperative Preparation and Postoperative Assessment

All patients had a positive Silfverskiold test. Patients usually had medial longitudinal arch collapse while standing with two feet parallel. The medial longitudinal arch of the feet restored when patients assumed a tiptoe position or were non-weight bearing. The included children were Viladot classification grade 3 and grade 4 with medial longitudinal arch collapse. Children with normal medial longitudinal arch with hindfoot valgus were excluded. All procedures were executed by a senior surgeon. We further excluded children with neuromuscular diseases, Sever’s apophysitis, joint hyperlaxity, and post-traumatic disorders.

A short leg walking cast was applied for three weeks for the healing of triceps surae from the Vulpius procedure. Plain radiographs of both feet were taken immediately following the operation to assure the implant was in good position. The follow-up radiographs, including anterior-posterior (AP) and lateral standing views of the foot, were taken at 6 weeks, 3 months, 6 months, 12 months, 18 months, and 24 months postoperatively. We removed implants in those children who wished to have their implants removed two years after the procedure.

We defined implant extrusion as the tip of the implant lateral to the middle third of the talus when weight bearing, based on an AP view of foot three months postoperatively, as shown in Figure 1.

Four cases were excluded because of immediate implant extrusion in postoperative radiograph. It was most likely due to technical error during the procedure. We also excluded a patient that implant was in situ in one foot and implant extrusion in the other foot.

### 2.3. Study Methods

During the study period, we included a total of 102 children who underwent the subtalar arthroereisis with the same result in both feet. Among them, there were 59 children with implant retained in both feet, while 43 children had implant extrusion. Two independent surgeons measured the talonavicular angle of the foot in AP view, testing the inter-observer correlation. Firstly, we measured radiographic data of the preoperative plain radiographs of both the in situ group (*n* = 59) and extrusion group (*n* = 43). Afterward, we added BMI data during the final follow-up as a point of a comparison between in the situ group and extrusion group. We examined the inter-observer correlation coefficient, Kolmogorov–Smirnov normality test, and Mann–Whitney U test for BMI data comparison.

Then, we measured the radiographic data of postoperative weight bearing radiographs taken during three-month postoperative outpatient visits. Similarly, we compared the radiographic data of implants between the in situ and extrusion group. Again, the Kolmogorov–Smirnov test was used to examine the normal distribution, the paired *t* test and Mann–Whitney U test were also used to compare the data.

For BMI data, patients with BMI ≥ 24 were defined as overweight (*n* = 28) and those with a BMI ≤ 18.5 as low body weight (*n* = 56). We compared the percentage of implant extrusion rate between both groups. Children with one foot extrusion were included in this part of the analysis. Medcalc^®^ Statistical Software version 17.9.7 (MedCalc Software, Mariakerke, Belgium) was used for data analysis, and *p* < 0.05 was defined as statistically significant.

The study had been approved by the Institutional Review Board of the National Taiwan University Hospital (201708084RIND).

## 3. Results

There were no statistically significant differences between the preoperative demographic data of both groups (i.e., in situ and extrusion), except preoperative talocalcaneal angle, calcaneal pitch, and talus inclination, as shown in Table 1 and Figure 2. The inter-observer correlation was excellent (correlation coefficient = 0.95), as shown in Figure 3. The average BMI of the implant in situ group was 19.5, whilst that of the implant extrusion group was 21.2 (*p* = 0.035), as shown in Figure 4. Demographic data of the overweight group (BMI ≥ 24) and the low body weight group (BMI ≤ 18.5) was shown in Table 2. There were 11 cases of bilateral extrusion in the overweight group (39%) and 13 cases of bilateral extrusion in the low body weight group (23%) (*p* = 0.0004), as shown in Table 3.

In view of 3 months after surgery, the postoperative radiographic data of the implant in situ group revealed calcaneal pitch, talonavicular angle, and Meary angle in both the AP and lateral view were better than the extrusion group (all *p* < 0.05). There was no significant difference in talus inclination angle between the two groups, as shown in Table 4 and Table 5, and Figure 5. In both groups, significant improvements were noticed in the postoperative radiographic angle compared to preoperative status. In general, the situ group is likely to have better surgical outcomes, as shown in Table 6.

## 4. Discussion

Symptomatic flexible flatfoot treatments include stretching exercises, shoe insoles, calcaneal stop procedures, extra-osseous subtalar arthroereisis, calcaneal osteotomy, and so on. Ozan et al. pointed out many children resist the arranged physical activities and insole treatment, leading to poor compliance [2]. In older children with severe deformity, calcaneal osteotomy may be one of the choices of treatment. Nonetheless, the calcaneal osteotomy is a more technically demanding surgery with a longer recovery period in comparison to extra-osseous subtalar arthroereisis surgery. In contrast, extra-osseous subtalar arthroereisis is relatively minor. By combining it with the Vulpius procedure and applying a short leg cast for both feet, the procedure can be done in a relatively short time. It is relatively safe surgery and only requires three weeks with a walking cast, which is easy for children to go through postoperatively.

According to Tong and Kong [3], medical longitudinal arch development of children stabilized between the age of seven and nine. Kubo et al. [4] indicated that the optimal time for a child to undergo subtalar screw arthroereisis is between the age of 9 and 12. Notably, the foot and foot arch at these ages are immature and still developing. Leaving room for further remodeling and development is of prime importance. Therefore, the foot arch could be in the proper position when the implant is removed two years after insertion. The implant removal is not likely to influence the nature of the foot and foot arch’s remodeling development, as described by Tong and Kong [3] and Vulcano et al. [5].

The radiographic outcomes of subtalar arthroereisis for pediatric flexible flatfoot have been reported [6]. Cao et al. reported 2 degree improvement in calcaneal pitch, 12 degree improvement in Meary’s angle from an AP view, and 15.4 degree improvement in Meary’s angle from a lateral view with use of the extra-articular subtalar implant Kalix II^®^ [7]. Faldini et al. further revealed that patient activity and life quality could be improved through corrected radiographic alignment with arthroereisis [8]. According to the previous literature, surgical complication rates have been reported up to 19% [9]. However, the consequences and the causes of implant extrusion have never been documented. Hence, our study aimed to focus on implant extrusion, and our analysis demonstrated that a higher BMI was related to subsequent implant extrusion during outpatient follow-up. In addition, our study further indicated that implant extrusion had a negative effect on angle correction, including the talar inclination angle from a lateral view, talocalcaneal and Meary’s angle in both the AP and lateral view. 

The effect of body weight on the medial foot longitudinal arch has been described. Woźniacka et al. [10] found a strong correlation between overweight and medial longitudinal arch pathology in 1115 children aged between 3 and 13 years. The most frequent pathology in the study is a high foot arch. Moreover, flatfoot was more frequently noted in the normal body weight group than in the overweight group. Our study provided comprehensive data to understand that overweight had a negative effect on correction of the foot arch [10]. Briefly, overweight led to implant extrusion in extra-osseous subtalar arthroereisis, which was evident in a lower correction rate of flatfoot. Since extraosseous implant could hold the position of the subtalar joint, overweight would have a higher possibility to cause implant extrusion, especially while walking.

Pavone reported a very high success rate of subtalar arthroereisis in 410 cases of flatfoot [11]. Only two (0.83%) implant extrusions were reported. However, their implants were screws with a specialized design for the insertion on the calcaneus. Drilling of the calcaneus is thus inevitable during the procedure [11]. Our study adopted an extra-osseous implant that did not require drilling calcaneus for fixation. In addition, all implant insertions could be completed with a 1.5 cm skin incision. We assumed that when dealing with flexible flatfoot, overweight children were more suitable for being treated using calcaneal screw [11,12]. In contrast, normal and underweight children were more suitable for extra-osseous subtalar arthroereisis. However, this requires further prospective randomized trials in the future.

Pavone et al. reported an improvement of 4 degrees in calcaneal pitch angle after a calcaneal stop procedure in which a screw was inserted by drilling the calcaneus [11]. In our series, although we had only a 1.5 degree improvement in the in-situ group and a 1.1 degree improvement in the extrusion group, we achieved a less invasive procedure with a non-bony procedure. Our data is comparable to the results from the previous extra-osseous subtalar arthroereisis [6]. Although there is not yet enough long term follow-up data, osteoarthritis is a concern in the subtalar arthroereisis procedure [1]. We believe a less bony procedure results in better protective effects on subtalar osteoarthritis.

Giannini et al. reported an improvement of 10 degrees in Meary’s angle in 21 children with flexible flatfoot [12]. Viladot et al. reported an improvement of 15 degrees in the talometatarsal 1 angle in 35 patients (37 feet) with adult flexible flatfoot [13]. However, both of them did not mention whether the implant extruded during the postoperative follow-up. In our study, the implant in situ group had an improvement of 7.8 degrees in Meary’s angle. In contrast, the implant extrusion group had a less improvement of 5.9 degrees in Meary’s angle.

Furthermore, we compared more detailed radiographic measurements including the talonavicular angle, Meary’s angle, talus inclination, talocalcaneal angle, and calcaneal pitch. It was evidenced that the correcting effect of the implant in situ group is better than that of the extrusion group. As shown in Table 5, the implant in situ group presents better surgical outcomes than the extrusion group. Bock et al. reported a high inter-observer reliability in the anteroposterior talonavicular angle, lateral talometatarsal 1 angle, and calcaneal pitch angle preoperatively and postoperatively [14]. We are thus convinced that our study further confirms the improvement of foot alignment based on Bock et al.’s recommended radiographic parameter.

To the best of our knowledge, there is no previous literature specifically focused on comparing the results of the implant between the in situ group and extrusion group. In our study, there was a statistical difference in correction between the implant in situ group and extrusion group. Both groups have also demonstrated a postoperative improvement in the foot alignment angle, although the angle corrected in the implant extrusion group performed worse than that in the in situ group.

Our study has several limitations. First of all, it is a retrospective study. In the future, a prospective study will answer some questions that we raised regarding the effects of preoperative BMI loss on surgical outcomes. Next, the evaluation on standing radiographs could be variable at times depending on the standing posture, although we have instructed radiology technicians for the standard techniques. In addition, the maturation of tarsal bones in this period of time might be another consideration. Meanwhile, our primary outcome focused on the relationship between implant extrusion and heavier body weight, and the difference in foot angle correction was focused on as a secondary outcome. The age distribution in extrusion group tended towards the older side, and older children tended to be heavier. Finally, our study could not investigate the radiographic outcomes after implant removal. However, based on over 20 years of experience, De Pellegrin et al. [15] considers subtalar extra-articular screw arthroereisis as an optimal technique for the correction of flatfoot.

## 5. Conclusions

In conclusion, our study has provided convincing evidence that overweight is associated with a higher incidence of implant extrusion after subtalar arthroereisis in a relatively large sample size of surgical cases. Postoperative radiographic angles were corrected in both the implant in situ group and the extrusion group. Nonetheless, the implant in situ group revealed better postoperative outcomes of Meary’s and the talonavicular angle from an anterior-posterior view, and the talar inclination angle from a lateral view. Further prospective study to investigate whether preoperative weight loss results in improved surgical outcomes in pediatric patients with flexible flatfoot is warranted in the future.

## Figures and Tables

**Figure 1 jcm-08-01273-f001:**
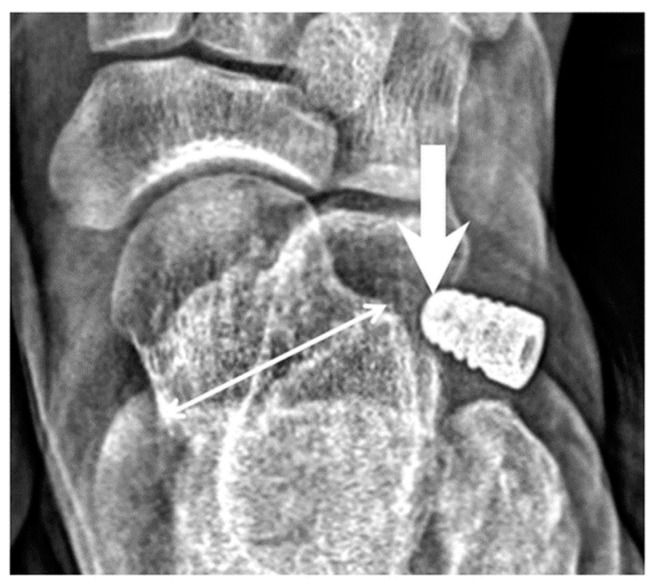
Anterior-posterior (AP) view of the foot with an implant (*n* = 204 feet of 102 patients); Arrow: tip of the implant. Thin arrow: talus width in AP view.

**Figure 2 jcm-08-01273-f002:**
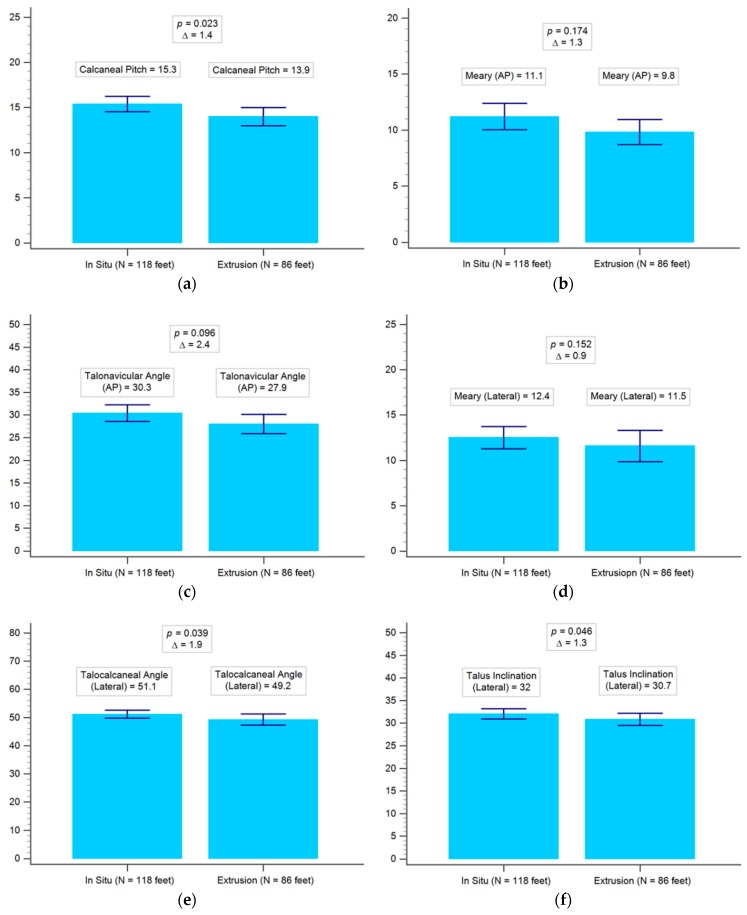
(**a**) Comparison of preoperative radiographic data for calcaneal pitch between the in situ and the extrusion group. (**b**) Comparison of preoperative radiographic data for anterior-posterior Meary angle between the in situ and the extrusion group. (**c**) Comparison of preoperative radiographic data for anterior-posterior talonavicular angle between the in situ and the extrusion group. (**d**) Comparison of preoperative radiographic data for lateral talonavicular angle between the in situ and the extrusion group. (**e**) Comparison of preoperative radiographic data for lateral talocalcaneal angle between the in situ and the extrusion group. (**f**) Comparison of preoperative radiographic data for lateral talus inclination between the in situ and the extrusion group. AP = anterior-posterior view.

**Figure 3 jcm-08-01273-f003:**
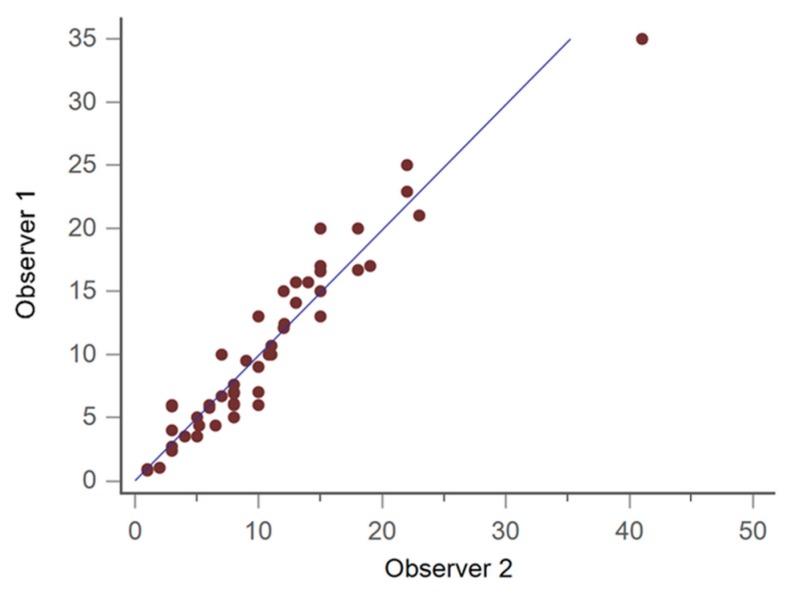
Inter-observer correlation test (20 feet of 10 patients, each observer).

**Figure 4 jcm-08-01273-f004:**
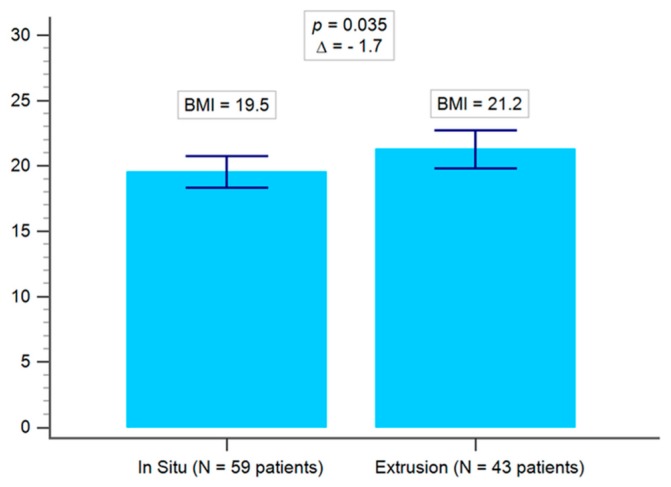
Comparison of mean body mass index (BMI) between the in situ and the extrusion group.

**Figure 5 jcm-08-01273-f005:**
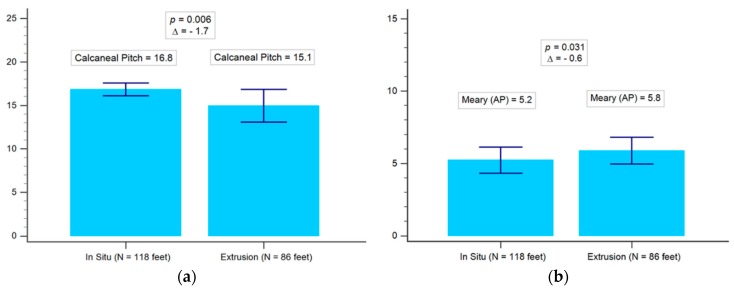
(**a**) Comparison of the postoperative radiographic data for anterior-posterior calcaneal pitch between the in situ and the extrusion group. (**b**) Comparison of the postoperative radiographic data for anterior-posterior Meary angle between the in situ and the extrusion group. (**c**) Comparison of the postoperative radiographic data for lateral talonavicular angle between the in situ and the extrusion group. (**d**) Comparison of the postoperative radiographic data for lateral Meary angle between the in situ and the extrusion group. (**e**) Comparison of the postoperative radiographic data for lateral talocalcaneal angle between the in situ and the extrusion group. (**f**) Comparison of the postoperative radiographic data for lateral talus inclination between the in situ and the extrusion group. AP = anterior-posterior view.

**Table 1 jcm-08-01273-t001:** Comparison of preoperative demographic data between the in situ and the extrusion group (*n* = 204 feet of 102 patients).

Demographic Data	In Situ	Extrusion	*p* Value
	(*n* = 118 Feet of 59 Patients)	(*n* = 86 Feet of 43 Patients)	
Age	8.6 ± 0.2	9.7 ± 0.3	0.021 *
Sex			
Male	42 (71%)	30 (70%)	
Female	17 (29%)	13 (30%)	
Calcaneal pitch	15.3	13.9	0.023 *
Meary (AP)	11.1	9.8	0.174
Talonavicular angle (AP)	30.3	27.9	0.096
Meary (lateral)	12.4	11.5	0.152
Talocalcaneal angle (lateral)	51.1	49.2	0.039 *
Talus inclination (lateral)	32.0	30.7	0.046 *

AP = anterior-posterior view. * *p* < 0.05.

**Table 2 jcm-08-01273-t002:** Comparison of preoperative demographic data between the overweight group (BMI ≥ 24) and the low body weight group (BMI ≤ 18.5) (*n* = 168 feet of 84 patients).

Demographic Data	Overweight (BMI ≥ 24)	Low Body Weight (BMI ≤ 18.5)	*p* Value
	(*n* = 56 Feet of 28 Patients)	(*n* = 112 Feet of 56 Patients)	
Age	10.1 ± 0.4	8.3 ± 0.2	0.0005 *
Sex			
Male	24 (86%)	35 (62.5%)	
Female	4 (14%)	21 (37.5%)	
Calcaneal pitch	13.7	15.2	0.018 *
Meary (AP)	9.1	11.4	0.019 *
Talonavicular angle (AP)	28.6	29.9	0.405
Meary (lateral)	9.5	12.4	0.002 *
Talocalcaneal angle (lateral)	53.1	51.7	0.004 *
Talus inclination (lateral)	29.7	32.6	0.005 *

AP = anterior-posterior view. * *p* < 0.05.

**Table 3 jcm-08-01273-t003:** Comparison of the demographic data between the overweight group and the low body weight group.

Demographic Data	Overweight (BMI ≥ 24)	Low Body Weight (BMI ≤ 18.5)
Age	10.1 ± 0.4	8.3 ± 0.2
Sex		
Male	24 (86%)	36 (63%)
Female	4 (14%)	21 (37%)
Both feet in situ	8 (29%)	32 (56%)
Both feet extrusion	11 (39%)	13 (23%)
One foot extrusion	9 (32%)	12 (21%)

**Table 4 jcm-08-01273-t004:** Comparison of the postoperative radiographic data between the in situ group and the extrusion group (*n* = 204 feet of 102 patients).

Radiographic Data	In Situ	Extrusion	*p* Value
	(*n* = 118 Feet of 59 Patients)	(*n* = 86 Feet of 43 Patients)	
Calcaneal pitch	16.8	15.1	0.006 *
Meary angle (AP)	5.2	5.8	0.031 *
Talonavicular angle (AP)	12.4	14.9	0.022 *
Meary angle (lateral)	4.8	5.9	0.038 *
Talocalcaneal angle (lateral)	47.2	45.0	0.019 *
Talus inclination (lateral)	25.9	25.4	0.632

AP = anterior-posterior view. * *p* < 0.05.

**Table 5 jcm-08-01273-t005:** Comparison of the preoperative and postoperative radiographic data between the in situ group and the extrusion group (*n* = 204 feet of 102 patients).

Radiographic Data	In Situ		*p* Value	Extrusion		*p* Value
	(*n* = 118 Feet of 59 Patients)			(*n* = 86 Feet of 43 Patients)		
	Pre-op	Post-op		Pre-op	Post-op	
Calcaneal pitch	15.3	16.8	<0.0001 *	13.9	15.1	<0.0006 *
Meary angle (AP)	11.1	5.2	<0.0001 *	9.8	5.8	<0.0001 *
Talonavicular angle (AP)	30.3	12.4	<0.0001 *	27.9	14.9	<0.0001 *
Meary angle (lateral)	12.4	4.8	<0.0001 *	11.5	5.9	<0.0001 *
Talocalcaneal angle (lateral)	51.1	47.2	<0.0001 *	49.2	45.0	<0.0001 *
Talus inclination (lateral)	32.0	25.9	<0.0001 *	30.7	25.4	<0.0001 *

AP = anterior-posterior view. * *p* < 0.05.

**Table 6 jcm-08-01273-t006:** Comparison of the radiographic improvement of the in situ and the extrusion group (*n* = 204 feet of 102 patients).

Radiographic Data	In Situ	Extrusion	*p* Value
	(*n* = 118 Feet of 59 Patients)	(*n* = 86 Feet of 43 Patients)	
	Improved Angle (Degree)	Improved Angle (Degree)	
Calcaneal pitch	2.7	2.6	0.732
Meary angle (AP)	7.8	5.9	0.008 *
Talonavicular angle (AP)	18.0	14.4	0.016 *
Meary angle (lateral)	7.9	7.0	0.096
Talocalcaneal angle (lateral)	5.1	5.5	0.355
Talus inclination (lateral)	7.3	5.5	0.005 *

AP = anterior-posterior view. * *p* < 0.05.

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
