# Peer review of "Body Weight Effects on Extra-Osseous Subtalar Arthroereisis"

_jcm, 2019, doi:10.3390/jcm8091273_

Round 1
Reviewer 1 Report
This manuscript compares the effect of BMI on implant extrusion after subtalar arthroereisis. The main question of the paper is interesting and worth to investigate. However this paper needs revision to improve its quality with regards to content and format.
Abstract: The abstract does not describe the essence of the paper well enough.
E.g. age of patients, in which period these data is collected (e.g. …between 12-2016 and 12-2018 120 patients had surgery due to flat feet…), how many patients were lost to followup is missing.
Detailed description of statistics (e.g.…Medcal was used …) does not belong in the abstract, but in the M&M-part.
2.1. patients criteria.
How many patients were operated? Is the data based on consecutive patients? How many patients are lost to follow up? Please add the month, not only the years “…from 2015 to 2017…”
2.2 Methods, why were patients with one foot implant in situ and the other foot with implant extrusion were excluded? This causes a bias.
4. Discussion
Line 188: The authors remove the implant strictly after 2 years, even if the foot is still developing? Why? Why not waiting until the foot is not developing any more, e.g. after the age of 13 or 14?
Line 186: the best age to perform an subtalar arthroereisis is between 9-12 years.
See:
„Outcome after subtalar screw arthroereisis in children with flexible flatfoot depends on time of treatment: Midterm results of 95 cases., Kubo H et al., J Orthop Sci. 2019”
This paper should be cited to underline the best time of treatment.
Overall a complication rate of 42% (43 cases extrusion of 102 collected cases) is very high. Why do the authors still recommend such a procedure? There are other procedures of subtalar arthroereisis, which are much more safe.
English language correction is recommended.
Author Response
Response to Reviewer 1 Comments
Point 1:
Abstract: The abstract does not describe the essence of the paper well enough. E.g. age of patients, in which period these data is collected (e.g. …between 12-2016 and 12-2018 120 patients had surgery due to flat feet…), how many patients were lost to followup is missing.
Detailed description of statistics (e.g.…Medcal was used …) does not belong in the abstract, but in the M&M-part.
Response 1: Please provide your response for Point 1. (in red)
- Thank you for the suggestion. We have revised the abstract as : " We collected data between May 2010 and Jan. 2017 with 59 cases of both feet implant in situ and 43 cases of both feet implant extrusion who had index surgery with Vulpius procedure. The average age of surgery of these 102 patients was 9 years. "
- We have deleted the excessive wording as suggested.
Point 2:
2.1. patients criteria.
How many patients were operated? Is the data based on consecutive patients? How many patients are lost to follow up? Please add the month, not only the years “…from 2015 to 2017…”
Response 2: Please provide your response for Point 2. (in red)
- We totally operated 145 number of patients. With the exclusion criteria, we studied 102 patients with bilateral operated feet. There was no loss of follow-up in our series in this retrospective study.
- We have indicated the month as suggested in our revised manuscript (Line 40-41, non track changes version).
Point 3:
2.2 Methods, why were patients with one foot implant in situ and the other foot with implant extrusion were excluded? This causes a bias.
Response 3:
- We agree; it seems that may cause bias. However, the study is on BMI and implant extrusion. Uneven cases may not represent well in our purpose of the study and discussion. In addition, the unilateral cases only count minority of cases. Therefore we excluded the unilateral involved patient in major part of our study.
Point 4:
Discussion
Line 188: The authors remove the implant strictly after 2 years, even if the foot is still developing? Why? Why not waiting until the foot is not developing any more, e.g. after the age of 13 or 14?
Response 4:
- It is a good question, thank you. We removed the implants after two years if patient and family requested as stated in our revised manuscript. It probably takes longer follow-up to answer the question if foot arch continues developing. With longer follow-up period of next stage, we can compare the group with/without implant removal on further study.
Point 5:
Line 186: the best age to perform an subtalar arthroereisis is between 9-12 years.
See:
„Outcome after subtalar screw arthroereisis in children with flexible flatfoot depends on time of treatment: Midterm results of 95 cases., Kubo H et al., J Orthop Sci. 2019” This paper should be cited to underline the best time of treatment.
Response 5:
- Yes, thank you for reminding. We have cited and discussed Kubo H. et al. work in the second paragraph of the discussion section, which underline the best time for a child to undergo subtalar screw arthroereisis is between 9 to 12 years of age (Line 172, non track changes version).
Point 6:
Overall a complication rate of 42% (43 cases extrusion of 102 collected cases) is very high. Why do the authors still recommend such a procedure? There are other procedures of subtalar arthroereisis, which are much more safe.
Response 6:
- It is an important question. We continue to use this procedure because of simplicity and minimal approach. It is an extra-osseous device without the need of bone drilling. We are now paying more attention to the security of implant fixation and carful casting technique to prevent from dislodging. The purpose of the study is to look at the BMI in relation to extrusion. We will be looking into other more secured device for heavier patient.
Point 7:
English language correction is recommended.
Response 7:
Yes, we have English edited to correct the language.
Reviewer 2 Report
It was a pleasure to review this paper.
The paper addresses an actual topic, results are well reported and properly discussed. However, some improvements are needed.
English form is poor, all the paper should be revised. I suggest to have an English native speaker reviewing the manuscript
Several references are missing all over the text
Better to change the term “film” with “X-Rays” all over the text.
Line 29: authors stated that device extrusion is a “common complication”: please add more details, haw common? Need references as well
Line 39: Why always Vulpius Procedure? Did you exclude patients whit accessory navicular ?
Line 40: How to detect a heel cord tightness?
Line 57: was the weight bearing allowed immediately? Please add more details
Line 163-164: references for all the surgical techniques described are needed
Line 167: I don’t believe calcaneal osteotomy to be a “complicated surgery”
Line 169: Calcaneal osteotomy could be performed, in some cases and according to patients requests, bilaterally
Line 170: English is not clear
Line 172: The sentence “a well-trained resident in the fourth year can be able to complete this procedure alone…” should be removed.
Line 190: Add references
Line 221 and line 275: please consider these papers in the discussion section:
Giannini S, Cadossi M, Mazzotti A, Persiani V, Tedesco G, Romagnoli M, Faldini C. Bioabsorbable Calcaneo-Stop Implant for the Treatment of Flexible Flatfoot: A Retrospective Cohort Study at a Minimum Follow-Up of 4 Years. J Foot Ankle Surg. 2017 Jul - Aug;56(4):776-782.
De Pellegrin M, Moharamzadeh D, Strobl WM, Biedermann R, Tschauner C, Wirth T. Subtalar extra-articular screw arthroereisis (SESA) for the treatment of flexible flatfoot in children. J Child Orthop. 2014 Dec;8(6):479-87. doi: 10.1007/s11832-014-0619-7. Epub 2014 Nov 21
Line 279: the term “handsome” seems quite strong. An English native speaker reviewing the manuscript would be helpful
Author Response
Response to Reviewer 2 Comments
Point 1:
English form is poor, all the paper should be revised. I suggest to have an English native speaker reviewing the manuscript
Response 1: Please provide your response for Point 1. (in red)
- Yes, we have English edited to correct the language.
Point 2:
Several references are missing all over the text
Response 2: Please provide your response for Point 1. (in red)
- Thank you for the suggestion. We have updated the list of references and make it more complete.
Point 3:
Better to change the term “film” with “X-Rays” all over the text.
Response 3:
- We have changed the term “film” to radiographs throughout the text.
Point 4:
Line 29: authors stated that device extrusion is a “common complication”: please add more details, haw common? Need references as well
Response 4:
- We have added reference by " Bernasconi, A.,Lintz, F.,Sadile, F. The role of arthroereisis of the subtalar joint for flatfoot in children and adults. EFORT Open Rev. Epub 2017/12/09; 2, 438-446. DOI : 10.1302/2058-5241.2.170009 " on the text (Line 29-30, non track changes version).
Point 5:
Line 39: Why always Vulpius Procedure? Did you exclude patients whit accessory navicular ?
Response 5:
- Tight Achilles tendon is a common problem in symptomatic flat foot. Those patient are unable to squat with foot flat on ground. They commonly perform the Silfverskiold test for indication of Vulpius procedure (Line 48, non track changes version).
- We excluded patients with accessory navicular. It is stated in exclusion criteria in patients criteria section ((Line 43-44, non track changes version).
Point 6:
Line 40: How to detect a heel cord tightness?
Response 6:
- We detect heel cord tightness by Silfverskiold test.
Point 7:
Line 57: was the weight bearing allowed immediately? Please add more details
Response 7:
- The bilateral short leg walking casts were applied for all patients and patients were allowed full weight bearing. The casts were removed in 3 weeks and continued full weight walking. The purpose of cast was to provide protection of Achilles tendon.
Point 8:
Line 163-164: references for all the surgical techniques described are needed
Response 8:
- Thank you for reminding, we have added reference by " Ozan, F., Dogar, F., Gencer, K., Koyuncu, S., Vatansever, F., Duygulu, F., Altay, T. Symptomatic flexible flatfoot in adults: subtalar arthroereisis. Ther Clin Risk Manag. Epub 2015/11/04. 11, 1597-602. DOI : 10.2147/TCRM.S90649" on the text (Line 161, non track changes version).
Point 9:
Line 167: I don’t believe calcaneal osteotomy to be a “complicated surgery”
Response 9:
-We have revised the statement as " But the calcaneal osteotomy is a more technical demanding surgery when compares to extra-osseous subtalar arthroereisis surgery,..." in the text (Line 164, non track changes version).
Point 10:
Line 169: Calcaneal osteotomy could be performed, in some cases and according to patients requests, bilaterally
Response 10:
- Yes, you are absolutely right, calcaneal osteotomy could be performed at the same time, however, it could bring some inconvenience to daily life and cause potential risk of non-union.
Point 11:
Line 170: English is not clear
Response 11:
- Sorry for the unclear statement. We have revised as "Compared to calcaneal osteotomy, extra-osseous subtalar arthroereisis is relatively minor." in the text (Line 165-166, non track changes version).
Point 12:
Line 172: The sentence “a well-trained resident in the fourth year can be able to complete this procedure alone…” should be removed.
Response 12:
- Agree, we have removed this sentence.
Point 13:
Line 190: Add references
Response 13:
- The reference is added as follow: " Tong, J. W., Kong, P. W. Medial Longitudinal Arch Development of Children Aged 7 to 9 Years: Longitudinal Investigation. Phys Ther. Epub 2016/02/20; 96, 1216-24. DOI : 10.2522/ptj.20150192" and " Vulcano, E., Maccario, C., Myerson, M. S. How to approach the pediatric flatfoot. World J Orthop. Epub 2016/01/26; 7, 1-7. DOI : 10.5312/wjo.v7.i1.1" on the text ((Line 170, non track changes version).
Point 14:
Line 221 and line 275: please consider these papers in the discussion section:
Giannini S, Cadossi M, Mazzotti A, Persiani V, Tedesco G, Romagnoli M, Faldini C. Bioabsorbable Calcaneo-Stop Implant for the Treatment of Flexible Flatfoot: A Retrospective Cohort Study at a Minimum Follow-Up of 4 Years. J Foot Ankle Surg. 2017 Jul - Aug;56(4):776-782.
De Pellegrin M, Moharamzadeh D, Strobl WM, Biedermann R, Tschauner C, Wirth T. Subtalar extra-articular screw arthroereisis (SESA) for the treatment of flexible flatfoot in children. J Child Orthop. 2014 Dec;8(6):479-87. doi: 10.1007/s11832-014-0619-7. Epub 2014 Nov 21
Response 14:
- Thank you for reminding. We have added the reference of "Giannini S, Cadossi M, Mazzotti A, Persiani V, Tedesco G, Romagnoli M, Faldini C. Bioabsorbable Calcaneo-Stop Implant for the Treatment of Flexible Flatfoot: A Retrospective Cohort Study at a Minimum Follow-Up of 4 Years. J Foot Ankle Surg. 2017 Jul - Aug;56(4):776-782", and "De Pellegrin M, Moharamzadeh D, Strobl WM, Biedermann R, Tschauner C, Wirth T. Subtalar extra-articular screw arthroereisis (SESA) for the treatment of flexible flatfoot in children. J Child Orthop. 2014 Dec;8(6):479-87. DOI: 10.1007/s11832-014-0619-7. Epub 2014 Nov 21. 8, 479-87." in the revised manuscript ((Line 207, 217 and 246, non track changes version).
Point 15:
Line 279: the term “handsome” seems quite strong. An English native speaker reviewing the manuscript would be helpful
Response 15:
- We agree, it is quite strong, the wording has been revised as "above average" in the text (Line 248, non track changes version).
- The manuscript is English edited.